# Thermal, Optical, and Microstructural Properties of Magnetron Sputter-Deposited CuSi Films for Application in Write-Once Blu-Ray Discs

**Feng-Min Lai [1], Yao-Tsung Yang [2] and Sin-Liang Ou [1,*]**

[1] Bachelor Program for Design and Materials for Medical Equipment and Devices, Da-Yeh University, Changhua 51591, Taiwan; fengmin@mail.dyu.edu.tw

[2] Department of Electrical Engineering, Da-Yeh University, Changhua 51591, Taiwan; tsung0712@hotmail.com

* Correspondence: slo@mail.dyu.edu.tw; Tel.: +886-4-8511888-2608

**Abstract:** In this study, 16-nm-thick CuSi films were deposited at room temperature by DC magnetron sputtering. The thermal, optical, and microstructural properties of CuSi films were investigated in detail. Moreover, the CuSi film was further used as a recording layer for write-once blu-ray disc (BD-R) applications. Based on the result of the reflectivity–temperature measurement, the CuSi layer had a decrease in the reflectivity between 180 and 290 °C. The as-deposited CuSi film possessed the $Cu_3Si$ phase. After annealing at 300 °C, the Si atoms existed in the CuSi film segregated and crystallized to the cubic Si phase. The activation energy of Si crystallization in the CuSi film was determined to be 1.2 eV. The dynamic tests presented that the BD-R containing the CuSi recording layer had minimum jitter values of 7.0% at 6 mW and 7.2% at 9 mW, respectively, for 1× and 4× recording speeds. This reveals that the CuSi film has great potential in BD-R applications.

**Keywords:** CuSi; write-once blu-ray disc; microstructure; crystallization kinetic; jitter value

## 1. Introduction

Recently, amorphous silicon (a-Si) has been used for the recording layer of write-once optical discs because of its simple fabrication process and low cost. It is well-known that materials with the crystallization temperatures lower than 500 °C are suitable for the recording films of optical discs. This is because the optical disc can be written efficiently with a relatively lower laser power of the laser pick up when the crystallization temperature of the recording layer is lower than 500 °C, which will extend the lifetime of the laser pick up. However, when the a-Si is prepared as the recording film, its extremely high crystallization temperature (700 °C) limits the application in the optical recording media, since a higher blue laser power is required in the recording process. This leads to an increase in the cost of the laser pick up. How to reduce the crystallization temperature of a-Si is very important. Among various techniques for reducing the crystallization temperature of a-Si, the metal induced crystallization (MIC) method is very useful [1–5]. As a result, various metals were employed to reduce the crystallization temperature of a-Si. In addition, various recording layers consisting of a-Si and metal have been proposed for optical recording media, especially for metal/a-Si bilayers. In 2004, the Cu/a-Si recording film was deposited for the write-once blu-ray disc (BD-R) [6]. The as-deposited Cu/a-Si film possessed the a-Si and Cu crystal structures. Then, the nucleation sites with the $Cu_3Si$ phase appeared at the process temperature of 100–250 °C. These nucleation sites can be used for the crystallization of a-Si. Finally, the crystallization temperature of a-Si was efficiently decreased to 485 °C due to the aid of $Cu_3Si$ nucleation sites. Later, a-Si combined with various metals, including Al, Ni, Mo, and Cu–Al alloy, were presented to prepare the Al/a-Si, Ni/a-Si, Mo/a-Si, and Cu-Al/a-Si recording films of BD-Rs [7–10].

Except for the metal/a-Si recording films, the other layer structure of the recording film applied for the BD-R was proposed in our previous research, i.e., the metal-silicide/a-Si bilayer films [11,12]. In 2011 and 2014, we presented the Si/CuSi and Si/NiSi bilayer recording films of BD-Rs, respectively [11,12]. Based on these two studies, the metal-silicide/a-Si bilayer had both a lower crystallization temperature of Si and a higher optical contrast in comparison to those of the metal/a-Si bilayer, resulting in the better recording performance of BD-R.

Actually, in addition to the metal/a-Si and metal-silicide/a-Si bilayer films, the metal-silicide recording layer also has high potential for BD-R applications. Our previous work also presented the NiSi alloy film as the recording layer of BD-R [13]. In commercial BD-Rs, the Cu/a-Si (or Si/CuSi) film is commonly used as the recording layer. In comparison to the Cu/a-Si bilayer, the Si/CuSi film possesses a much lower crystallization temperature of Si, which results in the better recording performance of the Si/CuSi BD-R and extends the lifetime of the laser pick up. However, when the BD-R is fabricated with the Cu/a-Si or the Si/CuSi recording layer, two materials (targets) are required for the films' deposition. In this study, to reduce the fabrication cost of the BD-R, the CuSi recording layer was prepared. As compared to the BD-R prepared with the Cu/a-Si or Si/CuSi recording layer, the CuSi BD-R can possess a simpler fabrication process since its recording film is single layer (not bilayer). CuSi alloy films with a thickness of 16 nm were deposited by DC magnetron sputtering. The composition of the CuSi layer was controlled to $Cu_{70}Si_{30}$ because of its lowest melt temperature of 802 °C in the Cu-Si phase diagram [14]. The microstructures before and after annealing were observed by transmission electron microscopy (TEM). Moreover, the recording characteristics of BD-R fabricated with the CuSi recording layer were also discussed.

## 2. Experimental Procedures

In this study, CuSi alloy thin films with a thickness of 16 nm were prepared on naturally oxidized Si, glass, and polycarbonate (PC) substrates by DC magnetron sputtering. The CuSi films were deposited at room temperature, and a $Cu_{70}Si_{30}$ composite target was employed during the sputtering process. In the sputtering equipment, a 3 inch DC planar magnetron, an Ar gas supply system, and a pumping system were attached to the vacuum chamber. The operational voltage and the sputtering power of the $Cu_{70}Si_{30}$ target were fixed at 400 V and 100 W, respectively. If the CuSi film is oxidized, the optical properties and recording characteristics of BD-R will be degraded. Therefore, to avoid the oxidation occurring in the CuSi recording layer, it should be sandwiched with the $ZnS–SiO_2$ protective layers. After pumping the chamber with the base pressure lower than $5\times10^{-7}$ Torr, the Ar gas was introduced and the working pressure was fixed at 5 mTorr during the growth process. Additionally, to investigate the crystallization mechanism of the CuSi film, some samples were subjected to the annealing processes in a furnace under a vacuum environment for 5 min and then quenched in ice water. For the fabrication of CuSi BD-Rs, the recording stacks were prepared on a 1.1-mm-thick PC substrate with the structure: Ag (95 nm)/$ZnS–SiO_2$ (35 nm)/CuSi (16 nm)/$ZnS–SiO_2$ (24 nm). When the dynamic tests were performed on the BD-Rs, the on groove recording method was used, and the track pitch between grooves in the BD-R was 0.32 μm. Finally, a PC transparent layer with a thickness of 0.1 mm was covered on the top of these layers by spin-coating. The layer structure of the CuSi BD-R is depicted in Figure 1. The films' thicknesses were measured using an alpha-step profile meter (Dektak 6M, Veeco, Plainview, NY, USA). Thus, the growth rates of various films were estimated, and the sample could be exactly made. Actually, various layer structures (including the films' thicknesses) of the CuSi BD-R samples were prepared, and we found that the CuSi BD-R with the optimum layer structure shown in Figure 1 possessed the better recording performance. Thermal properties of the CuSi film were measured by a home-made reflectivity–temperature analysis system at various heating rates of 5, 20, and 30 °C/min from room temperature to 500 °C. The sample for the thermal measurement was mounted on a Linkam THMS 600 heating stage (Linkam, Surrey, UK), and the measurement was performed in an argon protective atmosphere. The blue laser with a wavelength of 405 nm and a power of 15 mW was employed in this measurement, and the reflectivity change can

be recorded in real time when the process temperature was increased. The measurements of optical properties, including reflectivity, refractive index (n), and extinction coefficient (k) as a function of wavelength, were carried out using a UV–VIS–NIR spectrophotometer (Lambda900, PerkinElmer, Waltham, MA, USA) and a spectroscopic ellipsometer (SE 200, Sopra, CA, USA), respectively. The microstructures and crystallization mechanism of the CuSi films were characterized by TEM (JEM2100F, JEOL, Tokyo, Japan). The composition of the CuSi film was confirmed to be $Cu_{70}Si_{30}$ using an electron probe micro-analysis. The recording experiments consisting of jitter value and optical modulation for the CuSi BD-Rs were performed with a dynamic tester (ODU-1000, PULSTEC, Shizuoka, Japan). In the dynamic tests, the wavelength of 405 nm and the objective lens with a numerical aperture (NA) of 0.85 were employed to carry out the operation. The modulation code was (1, 7) RLL. The linear velocities of 1× and 4× recording speeds were 4.92 and 19.68 m/s, respectively.

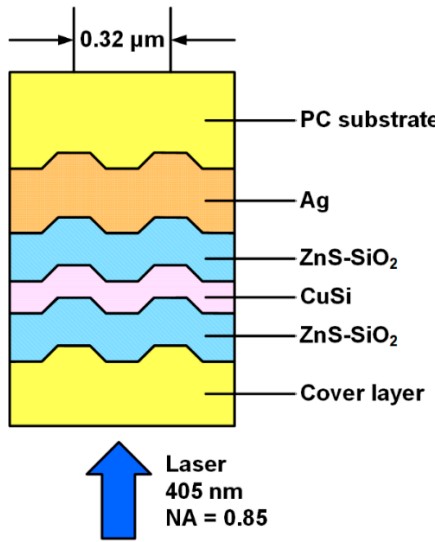

**Figure 1.** Layer structure of the write-once blu-ray disc (BD-R) fabricated with the 16-nm-thick CuSi recording layer. PC is polycarbonate. NA is numerical aperture.

## 3. Results and Discussion

Figure 2 shows the variations of reflectivity with the temperature for the 16-nm-thick CuSi layer at various heating rates of 5, 20, and 30 °C/min. The CuSi film exhibited an abrupt decrease in the reflectivity with increasing process temperatures between 180 and 256 °C, 206 and 282 °C, and 210 and 290 °C, as the heating rates were fixed at 5, 20, and 30 °C/min, respectively. The reflectivity change implied that the film had a structural transition, which could modify its optical characteristics [15]. According to the results, the annealed temperature of 300 °C was chosen to identify the microstructural change by TEM.

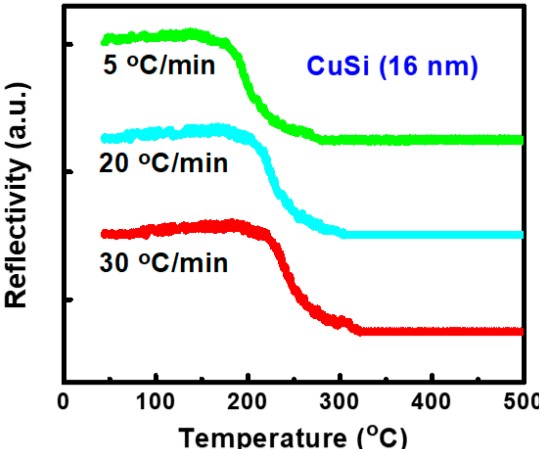

**Figure 2.** Relationship between reflectivity and temperature of the 16-nm-thick CuSi film measured at the heating rates of 5, 20, and 30 °C/min.

Figure 3a,b show the TEM images and diffraction patterns of the as-deposited and 300 °C-annealed CuSi films, respectively. In Figure 3a, the uniform and small grains with the size about 3–5 nm were observed in the as-deposited CuSi film. From the diffraction ring pattern, (320), (224), and (225) planes of the $Cu_3Si$ phase were analyzed. The diffraction rings of $Cu_3Si$ phase were identified by referring JCPDS-ICDD: 23-0224. Since the composition of the as-deposited CuSi film was $Cu_{70}Si_{30}$, it revealed that the Si atoms diffused into the $Cu_3Si$ matrix and caused a supersaturated $Cu_3Si$ crystalline structure during the film growth. After annealing at 300 °C for 5 min, it could be seen that a majority of this image still exhibited the uniform grains with the size of 5–10 nm; however, a few grains with the larger size of about 50–70 nm appeared during the annealing process. The diffraction ring pattern, as seen in Figure 3b, corresponded to the region circled with the red dash ring. Except for the existence of the $Cu_3Si$ phase, the cubic Si phase was crystallized with (111) orientation. Moreover, the selected area electron diffraction pattern of the larger grain marked with an arrow is shown in Figure 3c. From our calculation, this indicated that the larger grain had the cubic Si phase with $\langle 1\bar{1}0 \rangle$ zone axis. The diffraction rings and dots of the Si phase were identified by referring JCPDS-ICDD: 65-1060. The results demonstrated that Si atoms obtained enough energy to force themselves out of their original planes and were free to form the crystalline cubic Si in the $Cu_3Si$ nucleation sites as the film was annealed at 300 °C. The crystallization mechanism of the CuSi film was similar to that of the NiSi layer [13]. It is known that the crystallization temperature of pure a-Si is 700 °C, and it can be reduced to 485 °C by inserting a Cu layer. However, in this work, the crystallization temperature of Si was decreased below 300 °C efficiently by depositing the CuSi alloy layer.

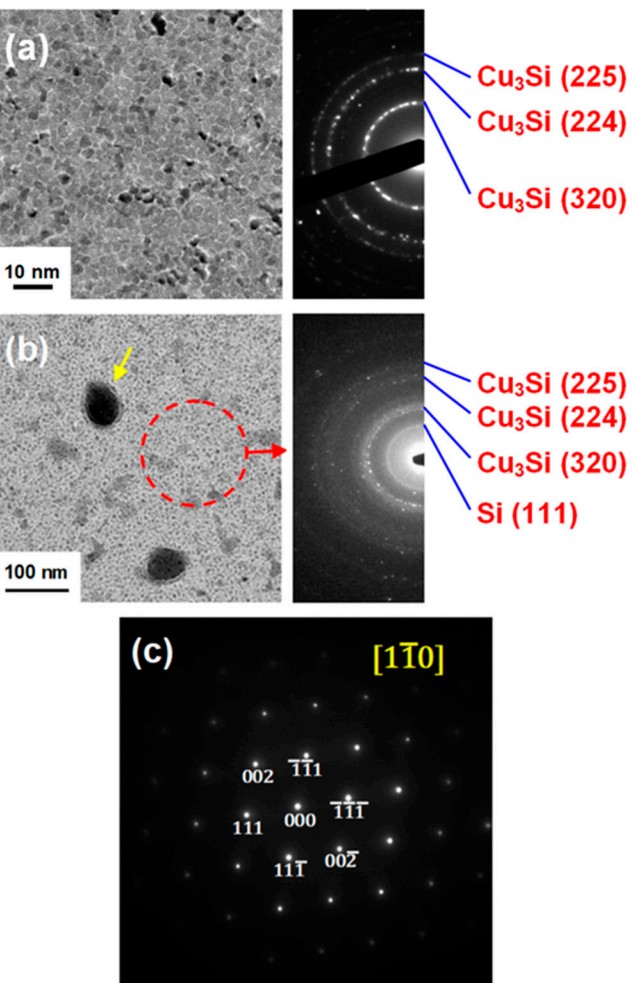

**Figure 3.** TEM bright field image and electron diffraction pattern of (**a**) as-deposited and (**b**) 300 °C-annealed CuSi (16 nm) films, and (**c**) the selected area electron diffraction pattern of the larger grain marked by an arrow in Figure 3b.

Based on the TEM results (Figure 3), we can realize that the variation in the reflectivity with an increment of the temperature (Figure 2) for the CuSi film contributed to the formation of Si crystallization. In Figure 2, the crystallization temperature ($T_x$) of Si was defined as the temperature at the midpoint of this broad reflectivity change, and these three $T_x$ were determined to be 218, 244, and 250 °C as the heating rates were 5, 20, and 30 °C/min, respectively. Due to less time for nucleation and crystallization at a high heating rate, it was expected that the crystallization temperature of Si would be increased with an increase of the heating rate. The crystallization temperatures with various heating rates can be related to the activation energy of crystallization by Kissinger's method [16]. The crystallization temperatures obtained at various heating rates were substituted into Kissinger's equation [16] to calculate the activation energy of crystallization. According to the Kissinger's equation,

$$\ln(A/T_x^2) = (-E_a/K_b) \times (1/T_x) + C \tag{1}$$

where $A$ is the heating rate, $T_x$ is the crystallization temperature, $K_b$ is the Boltzmann constant ($8.6 \times 10^5$ eV/K), $E_a$ is the activation energy for crystallization, and $C$ is a constant. The activation energy $E_a$ was evaluated by the slope of the $\ln(A/T_x^2)$ versus ($1/T_x$) curve, as shown in Figure 4. The activation energy of Si crystallization in the CuSi film was determined to be 1.2 eV. It was reported that the activation energy for the crystallization of pure amorphous Si is about 4.2 eV. Then, as a Cu layer was inserted to form the Cu/a-Si bilayer, the activation energy of Si crystallization decreased

to 3.3 eV [6]. Apparently, our work shows a much lower activation energy for the crystallization of Si by depositing the CuSi layer, indicating that a higher recording sensitivity can be achieved in the CuSi BD-R.

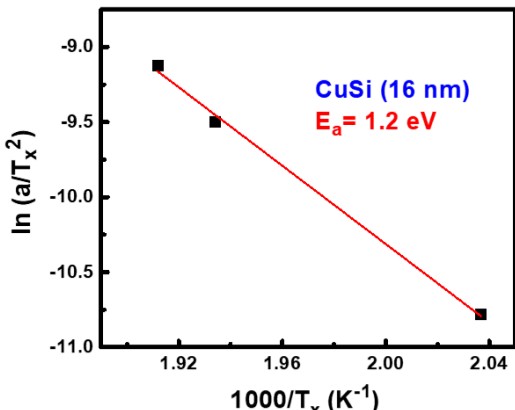

**Figure 4.** Plot of $\ln(A/T_x^2)$ versus $(1/T_x)$ of the CuSi film for the Si crystallization.

Optical properties of the as-deposited and 300 °C-annealed CuSi films were also studied. Figure 5a exhibits the *n* and *k* values of the CuSi (16 nm) film in the as-deposited and annealed states as a function of wavelength. We can observe that there were obvious differences between both *n* and *k* values before and after annealing at 300 °C. The as-deposited CuSi film had an *n* value (@405 nm) of 2.14, and it reduced to 1.45 (@405 nm) after annealing at 300 °C. Meanwhile, the *k* values (@405 nm) of the as-deposited and 300 °C-annealed CuSi films were 1.27 and 0.72, respectively. Figure 5b exhibits the variations of reflectivity with wavelengths of the as-deposited and 300 °C-annealed CuSi films. Under 405 nm wavelength, the reflectivities of the as-deposited and 300 °C-annealed CuSi films were 31.6% and 23.6%, respectively. The optical contrast was defined as $((R_1 - R_2)/R_1) \times 100\%$, where $R_1$ is the reflectivity of the as-deposited state and $R_2$ is the reflectivity of the annealed state. For the extraction of optical constants, the measured data were fitted with the Tauc–Lorentz model [17]. The corresponding optical contrast at the wavelength of 405 nm for the CuSi film was 25.3%, respectively. According to the experimental experience, the recording layer with an optical contrast higher than 15% had high potential for BD-R applications. From the TEM result (Figure 3), the obvious decrease in the reflectivity of the CuSi film after annealing at 300 °C could be ascribed to the formation of Si crystallization. The results reveal that the CuSi film was suitable for the recording layer of BD-R.

For the dynamic test, the jitter was defined as a measure of the spread in time for the transitions compared with their ideal position in the data stream. The jitter value means a distribution of rising or falling edge positions for all pits on the optical disc. In general, a low jitter value reveals a lower error as the marks are written in the optical disc. Based on the format of the optical disc, the jitter value can be expressed as follows [18]:

$$\text{Jitter value} = \sum_{k=1}^{n}(D_k - X_k) / \sum_{k=1}^{n} X_k \qquad (2)$$

where $D_k$ and $X_k$ are time data and frequency distribution, respectively. Figure 6 shows the dynamic test results of the BD-R fabricated with the 16-nm-thick CuSi recording layer that the jitter values and modulations vary with writing powers at 1× and 4× recording speeds. The suggested modulation and optimum jitter value for the BD-R should be larger than 0.4 and lower than 7.5%, respectively. It was found that the measured modulations at various writing powers and recording speeds for the CuSi BD-R were all higher than 0.4. As seen in Figure 3, the TEM observations indicates that the variations of structural phase and grain size uniformity in the as-deposited and annealed CuSi films resulted in high optical contrast. This led to the sufficient modulation of the BD-R before and after the laser

writing process. For the CuSi BD-R, the optimum jitter values were determined to be 7.0% at 6 mW and 7.2% at 9 mW, respectively, for 1× and 4× recording speeds. This work demonstrated that the CuSi recording film was highly feasible for the blue laser media applications.

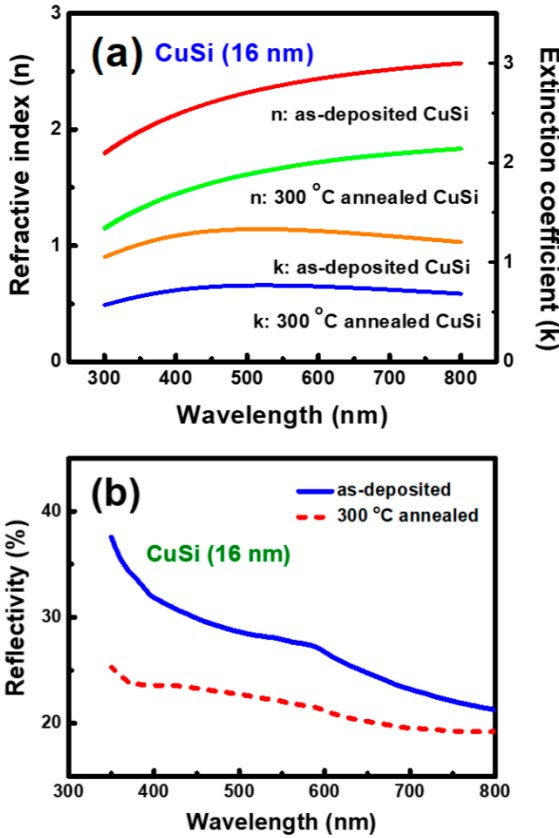

**Figure 5.** (**a**) Refractive index (*n*), extinction coefficient (*k*), and (**b**) reflectivity as a function of wavelength for the as-deposited and 300 °C-annealed CuSi films.

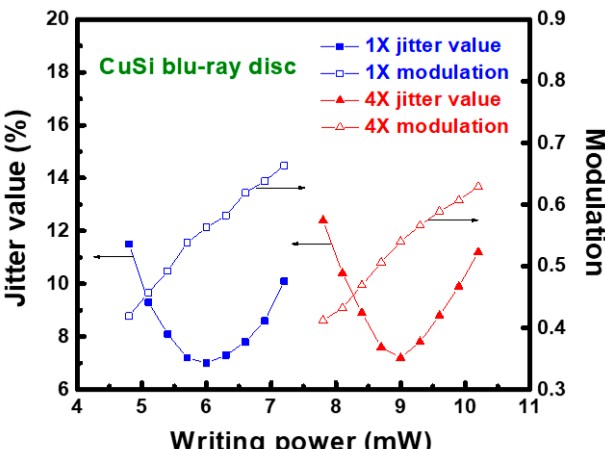

**Figure 6.** Jitter values and modulations as a function of writing power at 1× and 4× recording speeds for the BD-R fabricated with the 16-nm-thick CuSi recording film.

## 4. Conclusions

In summary, the thermal characteristics, microstructures, and optical properties of a 16-nm-thick CuSi film were investigated in detail. Moreover, the CuSi film was further used as the recording layer of the BD-R, and the recording performances were also analyzed. Based on the thermal properties and

TEM observations of the CuSi film, it was found that the crystallization temperature of a-Si for the CuSi film is as low as 180–290 °C. This reveals the crystallization temperature of a-Si can be reduced efficiently by depositing the CuSi film. After annealing at 300 °C, the Si crystallization was formed. In addition, the reflectivities (@405 nm) of the as-deposited and 300 °C-annealed CuSi films are 31.6% and 23.6%, respectively. This leads to the optical contrast (@405 nm) of 25.3% for the CuSi film. The results of dynamic tests show that the optimum jitter values of the CuSi BD-R are 7.0% at 6 mW and 7.2% at 9 mW, respectively, for 1× and 4× recording speeds. Our work reveals that the CuSi film is quite suitable for the BD-R applications.

**Author Contributions:** Conceptualization, F.-M.L., Y.-T.Y. and S.-L.O.; Methodology, F.-M.L. and Y.-T.Y.; Formal analysis, Y.-T.Y. and S.-L.O.; Investigation, Y.-T.Y. and S.-L.O.; Writing—original draft preparation, Y.-T.Y.; Writing—review and editing, F.-M.L. and S.-L.O.

**Funding:** The authors are thankful for the financial support of the Ministry of Science and Technology (Taiwan) under the Contract No. 107-2218-E-212 -001.

**Conflicts of Interest:** The authors declare no conflict of interest.

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
