# Peer review of "Thermal, Optical, and Microstructural Properties of Magnetron Sputter-Deposited CuSi Films for Application in Write-Once Blu-Ray Discs"

_coatings, doi:10.3390/coatings9040260_

Round 1
Reviewer 1 Report
1. Is the subject matter suitable for publication in COATINGS ?
Yes. The submission falls within the scope of the journal. The paper is very interesting. They are detailed enough to warrant a full paper.
2. What technological problems are addressed in the paper?
The present paper deals with the thermal, optical, and microstructural Properties of 2
Magnetron Sputter-Deposited CuSi Films for 3 Application in Write-Once Blu-Ray Disc.
3. Is the paper acceptable in its present form?
with minor corrections.
4. Is it clearly presented and well organized?
Yes.
5. Does it give adequate references related to work?
Yes.
6. Is the English satisfactory?
Yes.
7. Are the illustrations and tables all necessary and adequate?
In figure 3 TEM bright field image and electron diffraction pattern must be presented with higher definition , more enlightened and with more quality.
The PDF cards must be referred ( Cu3Si - PC Powder Diffraction Files, JCPDS-ICDD)].
8. Is the summary and conclusions adequate and informative?
Yes.
9. Are the conclusions sound and justified?
Yes
10. Further comments and specific suggestions:
The paper is very interesting.
General Remarks:
· Abstract: The aim of this work is not clearly stated.
· Please improve the introductory part of your paper in order to better highlight the motivation of your work.
· In figure 3 TEM bright field image and electron diffraction pattern must be presented with higher definition , more enlightened and with more quality. The PDF cards must be referred ( Cu3Si - PC Powder Diffraction Files, JCPDS-ICDD)].
Author Response
We appreciate the comments and suggestions from the reviewer 1, and the manuscript has been modified. In the revised manuscript, the added and modified texts are shown in red words. The detailed point-by-point responses to the reviewer’s comments are attached in the uploaded file.

Reviewer 2 Report
The paper presents new results for sputter-deposited CuSi films for application in blue-ray discs. There are several shortcomings which must be removed prior to publication.
Language (English) corrections required. The help of a native speaker is strongly recommended.
Line 37: what does BD-R exactly mean?
Experimental section: some details regarding the deposition system are required, e.g., magnetron and or target dimensions, operational voltage and power, etc. How exactly were the samples made? What are the reasons for the employed choice and thickness of layers (Figure 1)?
Line 80: laser power is missing
Line 137: equation should be on separate line
Figure 5: did you apply a model (which) for extraction of optical constants?
Line 166: jitter, can you provide a formula (on a separate line) regarding its definition?
Author Response
We appreciate the comments and suggestions from the reviewer 2, and the manuscript has been modified. In the revised manuscript, the added and modified texts are shown in red words. The detailed point-by-point responses to the reviewer’s comments are attached in the uploaded file.

Round 2
Reviewer 2 Report
The paper is improved and now ready for publication.